# ADAPTING CLIP FOR DETR-BASED OBJECT DETECTION

## ABSTRACT

Object detection involves class identification and spatial positioning. While DETR-based architectures have shown promising detection capabilities by framing the task as set prediction, prior approaches have limited refinement for object features, leading to inferior inherent understanding of objects, particularly when generalizing to unseen categories. To this end, we propose CLIP-DETR, a novel detection framework that harnesses the pretrained visual-linguistic capabilities of CLIP to enhance both the encoding and decoding processes in DETR models. Our method focuses on two key principles: 1) feature map sensitivity to objects, and 2) query adaptability. Extensive experiments demonstrate that CLIP-DETR significantly outperforms state-of-the-art models in object detection and open-vocabulary detection tasks, illustrating its superior generalization and recognition abilities.

## 1 INTRODUCTION

The evolution of object detection has seen significant advancements, beginning with the introduction of RCNN and YOLO variants Girshick et al. (2014); Ren et al. (2015); Redmon et al. (2016); Wang et al. (2021), which leveraged convolutional neural networks (ConvNets) to enhance accuracy and speed in object detection task. These methods, however, relied heavily on hand-designed components such as region proposal mechanisms and non-maximum suppresion, sometimes at the cost of computational efficiency. The advent of transformer-based models like DETR Carion et al. (2020), MaskFormerCheng et al. (2021), and Deformable DETR Zhu et al. (2020) revolutionized the field by introducing an end-to-end approach that eliminates the need for hand-crafted components, utilizing the transformer's ability to handle variable-sized inputs and model long-range dependencies. This shift towards transformers has led to state-of-the-art performances in dense prediction tasks, showcasing their flexibility and power in capturing complex spatial relationships and semantic information within image Liu et al. (2021); Li et al. (2022); Zong et al. (2023); Chen et al. (2023); Jia et al. (2023); Zhang et al. (2022); Li et al. (2023).

CLIP Radford et al. (2021) has revolutionized vision-language tasks by aligning images and text in a shared embedding space, enabling zero-shot recognition and generalized understanding across domains. Vision-language pretraining models like CLIP are increasingly being leveraged in object detection by incorporating semantic knowledge gained from large-scale image-text pairs. However, previous works that combine CLIP with object detection have several limitations. Most approaches only apply CLIP during the pretraining stage, without fine-tuning during the detection task, which limits the model's adaptability and precision. Furthermore, they focus primarily on label-based contrastive learning, omitting the critical role of spatial or scale information, which is essential for accurate localization. Additionally, CLIP's rich text embeddings are often underutilized in query generation and decoding, leading to less robust in challenging scenarios.

These shortcomings highlight the need for a more comprehensive approach that fully integrates CLIP's pretrained capabilities specifically into the object detection task. To address this gap, we propose CLIP-DETR, a novel framework that harnesses the visual-linguistic strengths of CLIP throughout both the encoding and decoding stages of detection.

At the core of CLIP-DETR encoding side is AlignNet, a label-aware and scale-aware feature refinement module to shape the semantics on the source feature map. AlignNet conducts a fine-grained alignment between image regions and both label concepts and object scale. It suggests a precise,

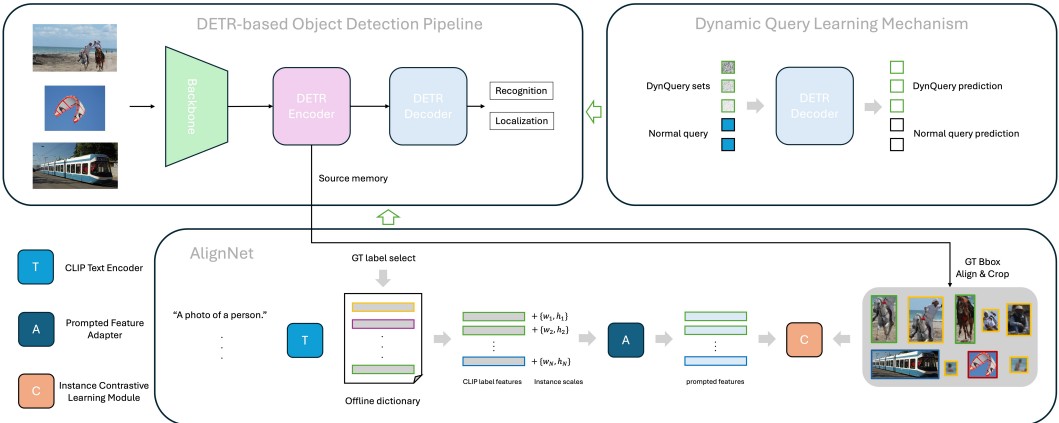

Figure 1: Overview of the CLIP-DETR architecture. The framework incorporates two key modules: AlignNet and the Dynamic Query Learning Mechanism (DynQL). AlignNet enhances the encoder by refining object representations with category- and scale-aware feature alignment, making the model more sensitive to object instances. DynQL dynamically adjusts query-object interactions, improving the decoder's robustness. Both modules are only applied during the training phase, ensuring that CLIP-DETR maintains the same inference-time computational efficiency as the baseline DETR architecture.

detailed correspondence between specific areas of an image and their associated labels, while also taking into account the size or scale of objects. This implies that the model not only recognizes what an object is (its label) but also refines its understanding based on the object's physical dimensions, enhancing both classification and localization. By applying contrastive learning during the fine-tuning stage, AlignNet directly boosts detection capabilities. Unlike previous approaches that rely on region proposals to pool object features Zhong et al. (2022); Wu et al. (2023c); Zareian et al. (2021), AlignNet leverages ground-truth bounding boxes (GT bboxes) to scale and crop object features from hierarchical levels, reducing noise from inaccurate proposals and enabling a cleaner, more efficient learning process.

Complementing AlignNet, CLIP-DETR also includes a dynamic query module called the Dynamic Query Learning Mechanism (DynQL). In DETR-based models, detection is formulated as a set prediction problem, followed by bipartite matching. The one-to-one matching mechanism struggles with finding suitable matches during early layers and training stages, as each query might not be able to find a suitable match within its receptive field, leading to less effective learning and potential underutilization of model capacity. While there existing works address the instability arising from bipartite matching by introducing denoising techniques Li et al. (2022); Zhang et al. (2022), our approach differs from them by seeing the decoding as process of linking queries with various informed degree to ground truth instances. In Fig.2, we visualize and compare the query predictions of DeformableDETR at the first and last decoder layers. It can be seen that for initial query closer to GT, its final prediction is more accurate, suggesting that queries starting closer to ground truth are easier to predict. Inspired by this observation, we introduce DynQL which is designed to improve the query-object linking process by introducing variability and robustness into the query learning process. DynQL applies the prompted object feature as the well informed queries, and applies multi-level noise to construct various initial distances to GT instances in the feature space. This dynamic grouping enhances the model's query sensitivity, allowing it to handle both close and distant queries with greater precision. Trained with various query-object pairs, the model learned to be more resilient to diverse object appearances and scales.

The contributions of this work are summarized as follows: (1) We introduce CLIP-DETR, a novel framework leveraging the pretrained CLIP to boost the training of DETR-based detectors. (2) We present AlignNet for enhanced category- and scale-aware feature refinement, and DynQL for robust query-object interaction modeling, leading to improved detection capacity, especially for unseen categories. (3) CLIP-DETR achieves state-of-the-art performance on both close-set object detection and open-vocabulary detection tasks.

## 2 RELATED WORK

### 2.1 OBJECT DETECTION TRAINING SCHEME

Co-DETR Zong et al. (2023) presents a novel collaborative hybrid assignments training scheme, where multiple parallel auxiliary heads are used during training, each supervised by one-to-many label assignments to enhance the encoder's learning capacity. However, the additional decoding heads significantly increase the computational load during training, making the process more time-consuming. $\mathcal{H}$-DETR Jia et al. (2023) and Group-DETR Chen et al. (2023) achieves faster convergence by introducing variants of one-to-many assignment. DAB-DETR Liu et al. (2021) formulate queries as 4D anchor boxes and dynamically refine them across decoder layers . DN-DETR Li et al. (2022) identifies instability in bipartite matching as another source of slow convergence and introduces a novel query denoising task by adding noise to 4D anchor boxes and class labels with a training objective of reconstructing the ground-truth ones . DINO Zhang et al. (2022) builds on DN-DETR and introduces a contrastive denoising training by introducing positive and negative queries by adding different scale of noise to ground truth boxes. However, the fixed scale of label noise limits the exploration of diverse query-object correspondences, as seen in real-world scenarios, which is particularly important when extending to open-set tasks. Cascade-DETR Ye et al. (2023) improves transformer-based detection methods by refining both the attention mechanism and query scoring, leading to better accuracy in complex environments. However, its real-world applicability is constrained by trade-offs in complexity, training time, and domain-specific performance. More recently, Rank-DETR Pu et al. (2024) focuses on resolving the misalignment between classification scores and localization accuracy, which undermines the quality of detection. It introduces a rank-oriented design to improve the selection of accurate bounding boxes. Nonetheless, Rank-DETR only addresses the misalignment between classification and localization tasks, without exploring the visual representation alignment in object detection.

### 2.2 VISUAL REPRESENTATION LEARNING FOR IMAGE REGIONS

Object detection fundamentally revolves around reasoning about image regions Everingham et al. (2010); Gupta et al. (2019); Krishna et al. (2017); Lin et al. (2014); Carion et al. (2020); Redmon et al. (2016); Wang et al. (2021); Ren et al. (2015); Tian et al. (2019). Most object detectors Zhu et al. (2020); Carion et al. (2020); Zhang et al. (2022); Liu et al. (2021); Li et al. (2022) are trained with supervision on the predictions from the decoder side, without specific emphasis on enhancing the quality of image region representations. To improve the learning process, semi-supervised learning methods have emerged Sohn et al. (2020); Xu et al. (2021); Zoph et al. (2020), utilizing pseudo-labels generated by teacher models to further train student detectors, thus reducing the reliance on extensive human annotations. Inspired by CLIP Radford et al. (2021), RegionCLIP Zhong et al. (2022) leverages vision-language pretraining to enhance region representations, allowing the model to recognize image regions with a large vocabulary. The concept of self-supervised learning is applied to region representation by encouraging the model to maximize the similarity between representations of different augmented views of the same image regions Hénaff et al. (2021); Ramanathan et al. (2021). To further reduce annotation costs, CLIPSelf Wu et al. (2023b) facilitates the transfer of CLIP's global vision-language alignment to local regions using self-distillation to avoid the direct association of individual regions with text. Different from these works, CLIP-DETR focuses on a more tailored refinement process specifically designed for the detector finetuning stage. By associating local image regions with both semantic- and scale-aware features, CLIP-DETR enables a more nuanced and accurate representation of real-world object variations. This dual-awareness of object class and scale allows for more precise reasoning about image regions, improving recognition and localization in a way that better reflects the complexities of natural scenes.

## 3 METHODOLOGY

### 3.1 PRELIMINARY

In a typical DETR-based object detection pipeline, the model begins by applying a pretrained image backbone to extract hierarchical image features from the input. These features are then passed

through an encoder, which refines them to produce a rich representation of the scene, often referred to as the source memory. This source memory serves as the key information pool for the decoder.

In the decoder, a set of trainable queries is used to interact with the source memory through cross-attention mechanisms, searching for object-related information. These queries iteratively refine the bounding box predictions and class scores by gathering relevant information from the source memory over multiple decoder layers. The bounding box prediction is progressively updated, becoming more accurate at each layer.

Finally, the predictions from the queries are matched to target objects or marked as background using a bipartite matching process. This process takes into account both the classification score and the distance and overlap between the predicted and ground-truth bounding boxes, ensuring an optimal assignment between predictions and objects.

## 3.2 ALIGNNET

In DETR-based object detection, the encoded feature map plays a crucial role as the source memory for the query-based decoding process. To accurately identify and localize objects, the feature map must provide a clear distinction between different instances. However, typical feature maps lack sufficient granularity, especially when handling instances with varying scales and categories. To address this limitation, we propose AlignNet, a module that enhances the encoded feature map by aligning it with both category-specific and object scale information, ensuring more fine-grained differentiation of object instances.

**Object encoded feature.** Given these multi-level ($L$) features $\mathcal{F}_i$ of dimensions $H_i \times W_i \times C$ outputted by the encoder, we perform ROI pooling using the ground truth bounding boxes $\mathcal{B}_{gt}$ on each feature level. This process generates a feature list of dimensions $L \times C$ for each object instance, where $L$ represents the number of feature levels, and $C$ is the feature dimension at each level. After pooling, we aggregate the information by averaging the features across the $L$ levels, producing a single object encoded feature $z_{enc}^i$ for each instance, as expressed in Eq.1:

$$z_{enc}^{i,l} = \mathbf{ROIPool}(\mathcal{F}^l, \mathcal{B}_{gt}^i) \in \mathbb{R}^{1 \times C}, \quad z_{enc}^i = \frac{1}{L} \times \sum_i^L z_{enc}^{i,l}, \tag{1}$$

where $i$ and $l$ indicate the ID of the instance in the image and the level number of the feature map, respectively. This encoded feature incorporates multi-scale information from all hierarchical feature levels, allowing the model to capture both fine-grained details from high-resolution features and more contextual information from coarser levels.

**Object attribute feature.** During training, with the ground truth labels and bounding box coordinates [cx,cy,w,h] available for each instance, we generate an object attribute feature that is both category- and scale-aware. First, we leverage the pretrained CLIP model by prompting its text encoder with "a photo of a [class]" to produce a label feature dictionary, where each class is associated with a corresponding feature $f_{cls}$. For each instance, we retrieve the corresponding class feature from this dictionary as the instance category feature $f_{cls}^i$. Next, to account for object scale, we concatenate the object's width and height [w,h] with the category feature, as we found through ablation studies that including just the width and height performed better than using the full bounding box information. As shown in Eq.2:

$$f_{cat}^i = \mathbf{Conca}(f_{cls}^i, [w, h]) \in \mathbb{R}^{1 \times (C_{clip}+2)}, \tag{2}$$

where $\mathbf{Conca}$ indicates the concatenation operation, and $f_{cat}^i$ represents the concatenated feature of the $i^{th}$ instance in the image. This concatenated feature is then passed through a linear layer to project it into the same embedding space as the encoded features, ensuring compatibility with the current task's feature space. As shown in Eq.3:

$$z_{attr}^i = \mathbf{Linear}(f_{cat}^i) \in \mathbb{R}^{1 \times C}, \tag{3}$$

where $z_{attr}^i$ indicates the final transformed attribute feature for the $i^{th}$ instance. This approach fully utilizes ground truth annotations without requiring additional human annotation for dense attributes.

**Alignment.** With both the object encoded feature $z_{enc}^i$ and the object attribute feature $z_{attr}^i$ in hand, we perform instance-wise contrastive learning between the two. We normalize the encoded

feature and attribute feature with L2 normalization to stabilize training and standardize features. We perform the dot product operation, scaled by a learnable logit $\beta$, to compute the pair-wise similarity between the two mode features within a batch, producing a similarity matrix $\mathcal{A}^{pred} \in \mathbb{R}^{N \times N}$. The target similarity matrix is an identity matrix $\mathcal{I}^{N} \in \mathbb{R}^{N \times N}$. The similarity score calculation between $i^{th}$ encoder feature and $j^{th}$ attribute feature, $\alpha_{i,j}^{pred}$, can be expressed as Eq. 4,

$$\alpha_{i,j}^{pred} = \beta \times \frac{z_{enc}^{i}}{\|z_{enc}^{i}\|_{2}} \cdot \frac{z_{attr}^{j}}{\left\|z_{attr}^{j}\right\|_{2}}, \tag{4}$$

where $\|.\|_{2}$ indicates the L2 normalization. Cross entropy loss, $\mathcal{L}_{CE}$, is applied along both modes axes to supervise the training of the similarity matrix, guiding the model in aligning latent feature representations more closely with label linguistic semantic and object scale as well as refine the inter-object relationships, as outlined in Eq. 5,

$$\mathcal{L}_{AlignNet} = \frac{\mathcal{L}_{CE}(\mathcal{A}^{pred}, \mathcal{I}^{N}, dim = 0) + \mathcal{L}_{CE}(\mathcal{A}^{pred}, \mathcal{I}^{N}, dim = 1)}{2}, \tag{5}$$

where $\mathcal{L}_{AlignNet}$ represents the AlignNet training loss. By bringing corresponding pairs closer together and pushing non-corresponding pairs apart, this contrastive learning process not only improves differentiation between different object classes but also captures variations in object size. This ensures more precise feature representation and enhanced object localization. With a more comprehensive attribute feature, AlignNet allows for a more nuanced and detailed contrastive learning step, reflecting real-world object variations and supporting more accurate object detection.

### 3.3 Dynamic Query Learning Mechanism

At the decoder side of DETR, a set of trainable queries is used to interact with the encoded source memory, extracting object-related information and performing object recognition and localization. Hence, the sensitivity to foreground object is important, particularly in challenging scenarios such as long-tail distributions, hard examples (e.g., small objects), and unseen object classes. We introduce a Dynamic Query Learning Mechanism (DynQL), which is designed to equip the model with a comprehensive perspective and understanding of the query decoding process and query-object correspondence.

**DynQuery content.** To creat varying initial query conditions, the DynQL introduces a spectrum of query sets, referred as DynQuery sets, ranging from less-informed to well-informed, based on the degree of object information embedded within each set, covering a broad wide of perspective of decoding starting point. This is achieved by utilizing the object prompt features as the fully informed queries and subsequently introducing varying levels of Gaussian noise to modulate the extent of information, as expressed in Eq. 6,

$$q_{DynQ-content}^{i,s} = \sqrt{\beta_{s}} \times \epsilon + \sqrt{1 - \beta_{s}} \times z_{attr}^{i} \in \mathbb{R}^{1 \times C}, \tag{6}$$

where $s$ indicates the $s^{th}$ DynQuery set, $\beta$ is a constant between 0 and 1 that controls the noise level, $q_{DynQ-content}^{i,s}$ is the $i^{th}$ DynQuery in the $s^{th}$ set, $\epsilon$ is the Gaussian noise and $\epsilon \sim \mathcal{N}(0, 1)$. Using the square root of $\beta$ ensures a smoother interpolation and provides a balanced and controllable way to introduce noise into the DynQuery sets, enabling effective exploration of the latent space while maintaining the influence of the original object proposal.

**DynQuery position.** For DynQL's positional encoding, we draw inspiration from DINO Zhang et al. (2022), incorporating random shifts and scales to GT positions of corresponding objects. The degree of shift and scale is controlled by parameter $\rho$, as expressed in Eq. 7,

$$p_{DynQ}^{i,s} = [p_{cent}^{i} + \lambda_{cent}^{i,s} \times \frac{\theta^{i}}{2}, p_{wh}^{i} + \lambda_{wh}^{i,s} \times \theta^{i}]; \quad \lambda \sim \mathcal{U}(0, \rho), \tag{7}$$

where $\lambda$ is the randomly picked degree in the uniform distribution $\mathcal{U}(0, \rho)$, $\theta$ indicates the width/height of the object, $p_{DynQ}$ is the obtained DynQuery position in the form of $[cx, cy, w, h]$, $p_{cent}$ and $p_{wh}$ are the ground truth object center and width/height. The DynQuery position are then encoded through positional encoding **PE** and a linear transformation to get the position encoding $q_{DynQ-pos}$, akin to conventional queries, as specified in Eq. 8,

$$q_{DynQ-pos}^{i,s} = \textbf{Linear}(\textbf{PE}(p_{DynQ}^{i,s})) \in \mathbb{R}^{1 \times C}. \tag{8}$$

Table 1: Object detection on COCO dataset.

| Method | Backbone | Epoch | #Query | AP | $AP_{50}$ | $AP_{75}$ | $AP_S$ | $AP_M$ | $AP_L$ |
|---|---|---|---|---|---|---|---|---|---|
| DETR Carion et al. (2020) | R50 | 500 | 100 | 42.0 | 62.4 | 44.2 | 20.5 | 45.8 | 61.1 |
| Conditional-DETR Meng et al. (2021) | R50 | 108 | 300 | 43.0 | 64.0 | 45.7 | 22.7 | 46.7 | 61.5 |
| Anchor-DETR Wang et al. (2022) | R50 | 50 | 300 | 42.1 | 63.1 | 44.9 | 22.3 | 46.2 | 60.0 |
| DAB-DETR Liu et al. (2021) | R50 | 50 | 900 | 45.7 | 66.2 | 49.0 | 26.1 | 49.4 | 63.1 |
| AdaMixer Gao et al. (2022) | R50 | 36 | 300 | 47.0 | 66.0 | 51.1 | 30.1 | 50.2 | 61.8 |
| DeformableDETR Zhu et al. (2020) | R50 | 50 | 300 | 46.9 | 65.6 | 51.0 | 29.6 | 50.1 | 61.6 |
| DAB-DeformableDETR Liu et al. (2021) | R50 | 50 | 300 | 46.8 | 66.0 | 50.4 | 29.1 | 49.8 | 62.3 |
| DN-DeformableDETR Li et al. (2022) | R50 | 50 | 300 | 48.6 | 67.4 | 52.7 | 31.0 | 52.0 | 63.7 |
| Dino-DeformableDETR † Zhang et al. (2022) | R50 | 12 | 900 | 49.4 | 66.9 | 53.8 | 32.3 | 52.5 | 63.9 |
| Group-DAB-DeformableDETR Chen et al. (2023) | R50 | 12 | 300 | 45.7 | - | - | 28.1 | 49.0 | 60.6 |
| $\mathcal{H}$-DeformableDETR Jia et al. (2023) | R50 | 12 | 300 | 48.7 | 66.4 | 52.9 | 31.2 | 51.5 | 63.5 |
| Co-DeformableDETR Zong et al. (2023) | R50 | 12 | 300 | 49.5 | 67.6 | 54.3 | 32.4 | 52.7 | 63.7 |
| Ours-DeformableDETR | R50 | 12 | 300 | 50.8 (+3.9) | 69.4 | 55.3 | 34.0 (+4.4) | 54.9 | 65.0 |
| Ours-DeformableDETR | CLIP-R50x64 | 12 | 300 | 52.0 (+5.1) | 71.1 | 56.6 | 34.6 (+5.0) | 56.3 | 66.7 |
| Dino-DeformableDETR † Zhang et al. (2022) | Swin-L | 36 | 900 | 58.5 | 77.0 | 64.1 | 41.5 | 62.3 | 74.0 |
| Group-Dino-DeformableDETR Chen et al. (2023) | Swin-L | 36 | 900 | 58.4 | - | - | 41.0 | 62.5 | 73.9 |
| $\mathcal{H}$-DeformableDETR Jia et al. (2023) | Swin-L | 36 | 900 | 57.9 | 76.8 | 63.6 | 42.4 | 61.9 | 73.4 |
| Co-DeformableDETR Zong et al. (2023) | Swin-L | 36 | 900 | 58.5 | 77.1 | 64.5 | 42.4 | 62.4 | 74.0 |
| Ours-DeformableDETR | Swin-L | 36 | 900 | **58.6** | 77.3 | 64.3 | **42.6** | 62.6 | 74.6 |

**DynQuery prediction.** Each set of DynQueries is processed in parallel with conventional queries within the decoder, allowing for a more comprehensive exploration of object-related clues within the source memory. To prevent information leakage between the different query sets, during the self-attention process, each DynQuery set can only interact with its own set and the conventional query set, while the conventional queries remain isolated from the DynQuery sets. The decoding process during training, incorporating both the conventional queries and DynQuery sets, can be formalized as Eq.9:

$$[\widetilde{y}, \widetilde{y}_{DynQ}] = \textbf{Decoder}([q, q_{DynQ-content}], \quad [q_{pos}, q_{DynQ-pos}], \quad \mathcal{F}_{[1...L]}), \tag{9}$$

where $\widetilde{y}$ and $\widetilde{y}_{DynQ}^s$ represent the predictions generated by the conventional queries and the Dyn-Query sets, respectively. Here, $q$ and $q_{pos}$ refer to the conventional queries and their corresponding positional encodings. After the prediction stage, each DynQuery is naturally matched with a ground truth instance, while conventional queries undergo a bipartite matching process to associate with either an instance or background. The same loss functions applied to conventional queries are also applied to DynQuery predictions, as expressed in the Eq.10:

$$\mathcal{L}_{DynQL} = \frac{\sum_s^S \sum_i^N \mathcal{L}_{conventional}(\widetilde{y}_{DynQ}^{s,i}, y^i)}{S \times N}, \tag{10}$$

where S and N represents the the number of DynQuery sets and the number of instances in the batch, respectively.

By integrating this dynamic query learning mechanism, the decoder gains a broader perspective and deeper understanding of query-object relationships, resulting in a model that is more robust and sensitive to the nuances of object detection.

## 4 EXPERIMENT

Our method is specifically designed for object detection tasks, and we begin by evaluating its performance on widely-used detection datasets, including COCO Lin et al. (2014) and LVIS Gupta et al. (2019). We then extend the evaluation to the open-vocabulary detection task, demonstrating its capability to generalize to unseen categories. Following these evaluations, we present ablation studies that explore the impact of different configurations of our method, providing insight into the contribution of each component.

### 4.1 OBJECT DETECTION

**Baselines.** For our object detection experiments, we selected Deformable-DETR Zhu et al. (2020), DINO Zhang et al. (2022), and Co-DETR Zong et al. (2023) as our baselines. Given that DINO, Co-DETR, and our CLIP-DETR are all training schemes designed for DETR-based object detectors,

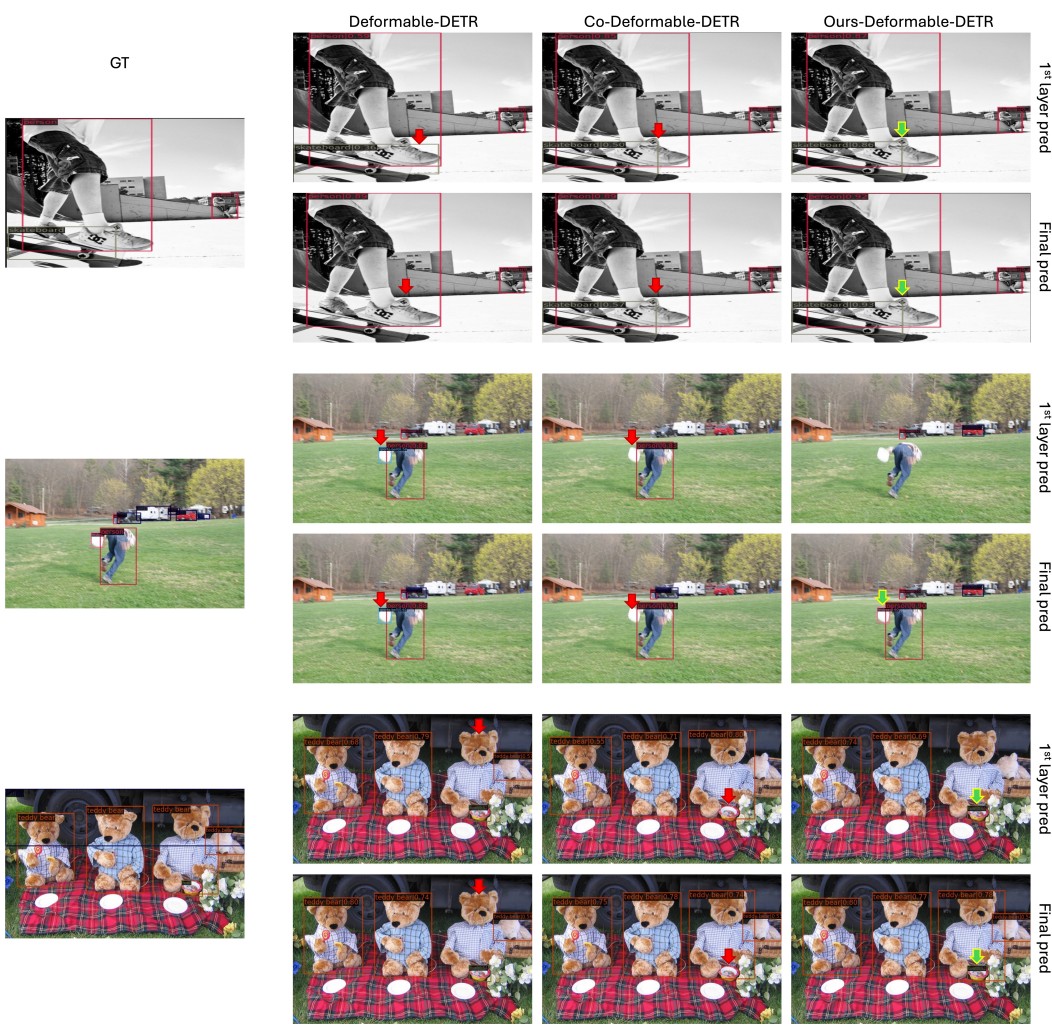

Figure 2: Visualization of detection results from the first and last decoder layers for Deformable-DETR, Co-DETR, and CLIP-DETR. The comparison illustrates that CLIP-DETR consistently enhances object detection starting from the first decoder layer. In cases where objects are missed early on, CLIP-DETR's query mechanism successfully identifies them in the final layer. This demonstrates CLIP-DETR's superior ability to refine and adapt queries throughout the decoding process, leading to improved overall detection performance.

Table 2: Object detection on LVIS dataset. † indicates training with LSJ augmentation.

| Detector | Training | Backbone | Epoch | #Query | AP | $AP_{50}$ | $AP_{75}$ | $AP_r$ | $AP_c$ | $AP_f$ |
|---|---|---|---|---|---|---|---|---|---|---|
| DeformableDETR | Co-DETR | R50 | 12 | 300 | 34.5 | 44.5 | 36.8 | 18.0 | 33.1 | 43.4 |
| | Ours | R50 | 12 | 300 | 35.6 (+1.1) | 45.7 | 37.6 | 19.2 (+1.2) | 33.7 | 44.1 |
| | Co-DETR † | R50 | 12 | 300 | 33.6 | 43.3 | 36.0 | 15.5 | 32.6 | 42.7 |
| | Ours † | R50 | 12 | 300 | 35.8 (+2.2) | 45.9 | 37.8 | 19.7 (+4.2) | 33.8 | 44.2 |

we chose Deformable-DETR as the foundational detector and built all models upon it to ensure a fair comparison.

**Setup.** We trained all models with a batch size of 16 and an initial learning rate of $2 \times 10^{-4}$. For the 12-epoch and 36-epoch training schedules, the learning rate was reduced by a factor of 0.1 at the $10^{th}$ and $30^{th}$ epochs, respectively. The label embeddings in CLIP-DETR were extracted from the

Table 3: Open-vocabulary object detection on coco dataset.

| Method | Base Detector | Backbone | Novel | Base | All |
|---|---|---|---|---|---|
| OVR-CNN Zareian et al. (2021) | Faster R-CNN | R50 | 22.8 | 46.0 | 39.9 |
| ViLD-text Gu et al. (2021) | Faster R-CNN | R50 | 5.9 | 61.8 | 47.2 |
| ViLD-image Gu et al. (2021) | Faster R-CNN | R50 | 24.1 | 34.2 | 31.6 |
| ViLD Gu et al. (2021) | Faster R-CNN | R50 | 27.6 | 59.5 | 51.3 |
| Detic Zhou et al. (2022) | Faster R-CNN | R50 | 27.8 | 47.1 | 45.0 |
| F-VLM Kuo et al. (2022) | Faster R-CNN | R50 | 28.0 | - | 39.6 |
| BARON-KD Wu et al. (2023a) | Faster R-CNN | R50 | 42.7 | 54.9 | 51.7 |
| RO-ViT Kim et al. (2023b) | Faster R-CNN | ViT-L/16 | 33.0 | - | 47.7 |
| CFM-ViT Kim et al. (2023a) | Faster R-CNN | ViT-L/16 | 34.1 | - | 46.0 |
| F-ViT-CLIPSelf Wu et al. (2023b) | F-ViT | ViT-L/14 | 44.3 | 64.1 | 59.0 |
| Prompt-OVD Song & Bang (2023) | Deformable DETR | ViT-B/16 | 30.6 | 63.5 | 54.9 |
| OV-DETR Zang et al. (2022) | Deformable DETR | R50 | 29.4 | 61.0 | 52.7 |
| Ours-OV-DETR | Deformable DETR | R50 | 30.8 (+1.4) | 61.5 | 53.5 |
| CORA Wu et al. (2023c) | DAB-DETR | R50 | 35.1 | 35.5 | 35.4 |
| Ours-CORA | DAB-DETR | R50 | 36.8 (+1.7) | 37.8 | 37.4 |
| CORA Wu et al. (2023c) | DAB-DETR | R50x4 | 41.7 | 44.5 | 43.8 |
| Ours-CORA | DAB-DETR | R50x4 | 42.5 (+0.8) | 46.2 | 44.9 |

text encoder of a pretrained CLIP model (RN50x64) in advance. We used mean Average Precision (mAP) as the primary metric to evaluate the detection performance.

**Results.** The object detection results on the COCO and LVIS datasets are shown in Table 1 and Table 2, respectively. From the second and third parts of Table 1, compared with other training schemes and Deformable-DETR variants, our CLIP-Deformable-DETR demonstrates the most significant performance improvements. With ResNet-50 backbone, ours achieves an mAP gain of 3.9% over the baseline. When using the CLIP image encoder as the backbone, our method provided an even larger improvement of 5.1% mAP over the baseline. On LVIS dataset, our method consistently outperformed Co-DETR across different training configurations and metrics, highlighting the generalization ability and stability of CLIP-DETR.

### 4.2 OPEN-VOCABULARY OBJECT DETECTION

**Baselines.** For the Open-vocabulary Object Detection task, we selected DETR-based models, OV-DETR Zang et al. (2022) and CORA Wu et al. (2023c), as our foundational detectors. We integrated our CLIP-DETR training scheme into these models to demonstrate its effectiveness in improving open-vocabulary detection performance.

**Setup.** We evaluate our approach on the widely used open-vocabulary detection benchmarks derived from COCO Lin et al. (2014). Following OVR-CNN Zareian et al. (2021), the COCO dataset is split into 48 base categories and 17 novel categories, with 15 categories removed due to lack of WordNet synsets. We refer to this benchmark as OV-COCO. For both baselines, we apply the same hyperparameter settings for training. We follow the standard practice for OV-COCO of reporting AP50 over the novel, base, and all classes.

**Results.** As shown in Table 3, CLIP-DETR consistently enhances the performance of the baselines across the dataset for both base and novel categories. Specifically, OV-DETR gains improvements of 1.4% and 0.5% on novel and base categories of the OV-COCO, respectively. CORA variants also experience gains of 1.7% and 0.8% on the novel categories. These results underscore the effectiveness and robustness of CLIP-DETR in handling open-vocabulary object detection.

### 4.3 ABLATION STUDY

We validate the effectiveness of each component of CLIP-DETR, including AlignNet and DynQL, in enhancing detection performance. Additionally, we explore the impact of varying configurations

Table 4: Effectiveness of each component.

| Method | AP | $AP_S$ | $AP_M$ | $AP_L$ |
|---|---|---|---|---|
| baseline | 46.3 | 29.5 | 49.4 | 61.1 |
| + AlignNet | 49.8 | 33.7 | 54.1 | 63.9 |
| + DynQL | 49.3 | 33.2 | 53.6 | 64.3 |
| + CLIP-DETR | **50.8** | **34.0** | **54.9** | **65.0** |

Table 5: Category-aware and scale-aware feature refinement of AlignNet.

| Method | Feature | AP | $AP_S$ | $AP_M$ | $AP_L$ |
|---|---|---|---|---|---|
| baseline | - | 46.3 | 29.5 | 49.4 | 61.1 |
| + CLIP-DETR | label | 46.9 | 32.3 | 50.8 | 60.2 |
| | label + bbox | 48.5 | 32.9 | 52.8 | 62.4 |
| | label + scale | **50.8** | **34.0** | **54.9** | **65.0** |

of CLIP-DETR on model performance, providing insights for future research. All ablation studies are conducted on the COCO detection task using Deformable DETR with a 12-epoch training setup as the baseline.

**Effectiveness of each component.** We design experiments to rigorously assess the contributions of the AlignNet and the DynQL, as detailed in Table.4. The results in the second and third lines reveal that integrating either AlignNet or DynQL with the baseline model independently results in substantial performance enhancements. Moreover, when the baseline is augmented with the full suite of CLIP-DETR components, i.e. both AlignNet and DynQL, we observe even more pronounced improvements. These findings unequivocally demonstrate the significant impact of both modules, underscoring their individual and combined strengths in bolstering the model's performance. This evidence firmly establishes the vital roles AlignNet and DynQL play in achieving the superior capabilities of CLIP-DETR.

**Category-aware and scale-aware feature refinement of AlignNet.** Humans naturally recognize objects by being aware of both their labels and sizes. To investigate whether this logic holds for object detection models, we conducted an ablation study comparing different strategies for refining the encoded source memory with category- and size-aware features. Specifically, we compared the traditional CLIP-based contrastive learning approach, which focuses solely on category-based region-text alignment, with our proposed contrastive learning that incorporates object scale information. We tested three configurations:

- The object feature is paired with label text embedding.
- The object feature is paired with label text embedding concatenated with bounding box coordinates [cx,cy,w,h].
- The object feature is paired with label text embedding concatenated only with object scale [w,h].

We utilize label features extracted from the CLIP-RN50x64 text encoder as label embeddings for all experiments. As shown in Table 5, adding object alignment during detection training significantly improves the model's performance across all configurations. Notably, the best results are achieved when the model is aligned using both label and scale information, confirming that our feature encoding method aligns more closely with the core logic of object detection tasks. **Analysis.** Aligning only with the object's width and height [w,h] outperformed the full bounding box alignment [cx,cy,w,h]. This can be attributed to the fact that, during training, the anchor can accurately regress to the correct object center using scale information and its current position. However, embedding [cx,cy] into the feature may confuse the model, as images are inherently translation-invariant, and [cx,cy] might not be suitable for refining the source memory encoding. This result suggests that scale information is more relevant than absolute position for effective feature alignment in detection tasks.

**Various Informed DynQuery Sets.** We explored how different levels of noise in the DynQuery sets influence model performance by keeping the position scale as $\rho = 1$. First, by setting the

Table 6: Ablation on the range of $\beta$ for the construction of various informed DynQuery Sets.

| Method | $\beta$ | AP | $AP_S$ | $AP_M$ | $AP_L$ |
|---|---|---|---|---|---|
| baseline | - | 46.3 | 29.5 | 49.4 | 61.1 |
| + CLIP-DETR | $[0.3] \times 5$ | 47.8 | 32.7 | 52.1 | 61.1 |
| | $[0.5] \times 5$ | 48.2 | 32.3 | 52.5 | 61.8 |
| | $[0.9] \times 5$ | 47.7 | 31.9 | 52.0 | 61.4 |
| | $[0.1, 0.3, 0.5, 0.7, 0.9]$ | **50.8** | **34.0** | **54.9** | **65.0** |

Table 7: Ablation on the number of DynQuery sets.

| Method | $\beta$ | AP | $AP_S$ | $AP_M$ | $AP_L$ |
|---|---|---|---|---|---|
| baseline | - | 46.3 | 29.5 | 49.4 | 61.1 |
| + CLIP-DETR | $[0.3] \times 1$ | 46.8 | 30.1 | 51.1 | 60.9 |
| | $[0.5] \times 1$ | 47.4 | 30.9 | 50.6 | 61.7 |
| | $[0.9] \times 1$ | 46.7 | 31.0 | 50.7 | 60.9 |
| | $[0.1, 0.3, 0.5, 0.7, 0.9]$ | **50.8** | **34.0** | **54.9** | **65.0** |
| | $[0.1, 0.3, 0.5, 0.7, 0.9] \times 2$ | 48.4 | 32.8 | 52.2 | 62.1 |

number of DynQuery sets to 5, we tested performance across various noise levels: 0.3, 0.5, and 0.9, as well as a gradual increase from 0.1 to 0.9, as shown in Table 6. The results indicate that using a uniform distribution of noise levels yields the best performance. This suggests that covering a broad spectrum of query conditions, from minimally informed to highly informed, enhances the decoder's adaptability and overall understanding of query-object interactions. This diversity in noise strengthens the decoder's capacity to access relevant object information across different query scenarios, improving its object detection performance. Next, we investigated the impact of varying the number of DynQuery sets, as shown in the Table 7. When increasing the number of sets to 10 or reducing it to fewer than 5, we observed a decline in performance compared to using 5 sets. We attribute this to overfitting when too many query sets are introduced, as the decoder becomes overly reliant on prompted queries. Conversely, using fewer than 5 sets results in insufficient coverage of diverse query conditions, limiting the model's ability to generalize effectively. Thus, the choice of 5 query sets provides an optimal balance, offering the necessary diversity without compromising performance.

## 5 CONCLUSION

In this paper, we present CLIP-DETR, a novel framework that integrates the visual-linguistic knowledge of the pretrained CLIP model into the DETR-based object detection pipeline. By leveraging CLIP's capabilities, we introduced two key modules—AlignNet and DynQL. AlignNet enhances the encoder representation by refining it with category- and scale-aware object features, and DynQL equips the decoder with more flexible and robust query-object interaction logic. Extensive experiments demonstrate that CLIP-DETR consistently outperforms existing state-of-the-art models on both traditional object detection and open-vocabulary detection tasks, proving the effectiveness and generalization ability of our approach. Moving forward, our method opens up new possibilities for leveraging pretrained vision-language models in detection tasks, inspiring future work in object detection frameworks that can effectively bridge the gap between feature extraction and query-based detection.

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

# A APPENDIX

## A.1 DISCRIMINATIVE SCORE

The discriminative score in CLIP-DETR offers a quantifiable measure of the model's ability to distinguish foreground objects from the background within encoded feature maps. By leveraging the l2-norm of feature vectors at each spatial coordinate of the encoder's output ($C \times H \times W$), we derive a discriminability score map ($1 \times H \times W$). This score map serves as a direct indicator of the model's efficacy, with higher scores in specific areas correlating with improved object detection capabilities. Such a method allows for an intuitive assessment of the encoding process's success in enhancing feature discriminability, critical for subsequent decoding stages.

## A.2 DYNQUERY POSITION SHIFTING AND SCALING.

We investigated the impact of different scales $\rho$ on shifting and scaling the GT position for DynQuery position. The results in Tab. 8 indicate that $\rho = 1$ achieves the best performance. Scales that are too small or too large fail to adequately challenge the decoder or overwhelm the decoder's attention mechanism, hindering its ability to learn the relationship between objects' real position and queries effectively.

Table 8: Ablation on Position Scale $\rho$ Value.

| Method | $\rho$ | AP | $AP_S$ | $AP_M$ | $AP_L$ |
|---|---|---|---|---|---|
| baseline | - | 46.3 | 29.5 | 49.4 | 61.1 |
| | 0.1 | 45.9 | 30.3 | 49.1 | 58.8 |
| | 0.4 | 47.6 | 30.2 | 50.5 | 62.8 |
| + CLIP-DETR | 0.8 | 49.4 | 33.5 | 53.4 | 63.5 |
| | 1.0 | **50.8** | **34.0** | **54.9** | **65.0** |
| | 1.5 | 48.8 | 32.4 | 53.1 | 62.7 |

## A.3 DECODER LAYER-WISE PERFORMANCE

In Tab.9, we present a comparative analysis of the detection performance across all decoder layers for three methods: Deformable-DETR, Deformable-DETR enhanced with Co-DETR, and Deformable-DETR enhanced with CLIP-DETR. The results clearly demonstrate the impact of CLIP-DETR, particularly in the early layers of the decoder.

Notably, CLIP-DETR achieves an AP of 46.2 at the first decoder layer, which significantly outperforms both Deformable-DETR and Co-DETR. This early-stage improvement suggests that CLIP-DETR equips the model with a stronger initial understanding of query-object relationships, allowing it to identify objects more accurately from the outset. As the layers progress, CLIP-DETR continues to outperform the other methods, achieving consistent gains across all metrics.

This improvement highlights the ability of CLIP-DETR to refine the decoder's comprehension over successive layers, ensuring that even if certain objects are missed in earlier layers, the model can successfully detect them in later ones. The results underscore the effectiveness of integrating CLIP's pretrained knowledge into both the encoder and decoder, improving the overall decoding process for object detection tasks.

Table 9: Performance comparison across layers.

| Method | Decoder Layer | AP | $AP_{50}$ | $AP_{75}$ | $AP_S$ | $AP_M$ | $AP_L$ |
|---|---|---|---|---|---|---|---|
| Deformable-DETR | 1 | 39.1 | 55.0 | 42.8 | 24.7 | 43.0 | 49.4 |
| | 2 | 43.3 | 60.4 | 47.1 | 26.9 | 46.7 | 55.7 |
| | 3 | 45.1 | 62.4 | 49.3 | 27.9 | 48.3 | 58.4 |
| | 4 | 45.9 | 63.5 | 50.1 | 28.9 | 49.2 | 60.1 |
| | 5 | 46.3 | 64.1 | 50.4 | 29.1 | 49.5 | 60.7 |
| | 6 | 46.3 | 64.3 | 50.5 | 29.5 | 49.4 | 61.1 |
| +Co-DETR | 1 | 41.7 | 57.5 | 45.8 | 27.3 | 45.2 | 53.9 |
| | 2 | 46.9 | 64.4 | 51.3 | 30.1 | 50.9 | 60.3 |
| | 3 | 48.5 | 66.2 | 53.0 | 31.3 | 52.3 | 62.9 |
| | 4 | 48.9 | 66.9 | 53.4 | 31.7 | 52.5 | 63.8 |
| | 5 | 49.0 | 67.0 | 53.4 | 31.8 | 52.4 | 64.0 |
| | 6 | 49.5 | 67.6 | 54.3 | 32.4 | 52.7 | 63.7 |
| +CLIP-DETR | 1 | 46.2 | 63.9 | 50.7 | 30.6 | 50.9 | 58.7 |
| | 2 | 50.0 | 68.6 | 54.6 | 33.2 | 54.2 | 64.0 |
| | 3 | 51.3 | 70.2 | 56.0 | 34.2 | 55.7 | 65.5 |
| | 4 | 52.0 | 71.0 | 56.6 | 34.9 | 56.4 | 66.4 |
| | 5 | 52.2 | 71.3 | 56.7 | 34.6 | 56.4 | 67.0 |
| | 6 | 52.1 | 71.1 | 56.6 | 34.6 | 56.3 | 66.7 |

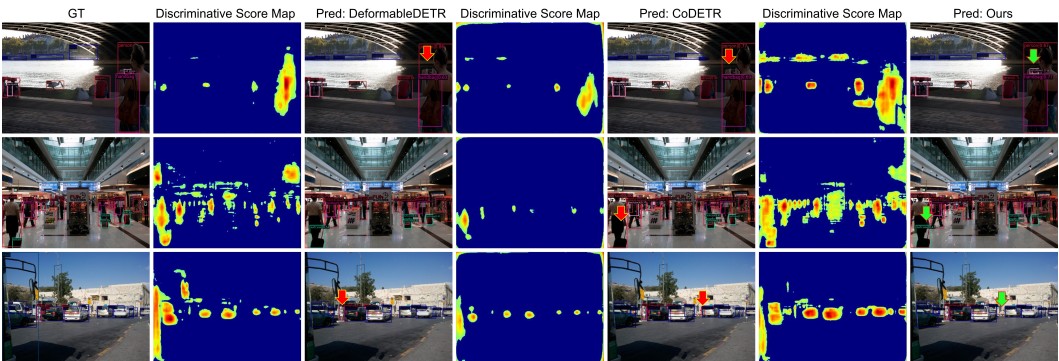

Figure 3: Visual Comparison of Discriminative Scores and Detection Results: This figure demonstrates the discriminative score map and detection results from Deformable-DETRZhu et al. (2020), CoDETRZong et al. (2023), and CLIP-DETR, with both CoDETR and CLIP-DETR building upon the Deformable-DETR. Following the approach outlined in CoDETR, we visualize the discriminative score map to highlight the encoding capabilities of each model. The superiors of CLIP-DETR are highlighted by green arrows, indicating its better performance, while the inferiors of the other baselines are marked with red arrows. Areas with a discriminative score below 0.5 are intentionally omitted to focus on more distinct regions. CLIP-DETR is shown to produce more distinguishable encoded feature maps and more accurate detection results.

