# OpenReview forum: "Adapting CLIP for DETR-based Object Detection"
_ICLR.cc/2025/Conference — Submitted to ICLR 2025_

### Official Review · Reviewer_Rz2L · 2024-11-01

**Soundness:** 3
**Presentation:** 2
**Contribution:** 2
**Rating:** 5
**Confidence:** 2

**Summary:**

This paper proposes a versatile, high-performance framework that leverages pretrained CLIP to enhance DETR-based object detection through category- and scale-aware contrastive learning. Additionally, a dynamic query mechanism in the decoder stabilizes and improves bounding box prediction, resulting in more robust and accurate detection outcomes.

**Strengths:**

1. The authors introduce a novel, generalized training framework for DETR, achieving notable accuracy improvements in both traditional object detection and open-vocabulary tasks, showcasing the framework’s versatility and effectiveness.

2. By incorporating object scale features into the CLIP alignment, the model captures richer physical and structural information, enhancing its capacity to recognize objects with diverse characteristics.

3. The introduction of a Dynamic Query mechanism in the decoder increases the robustness and stability of bounding box predictions, leading to more accurate and reliable detection results.

**Weaknesses:**

1. The paper lacks a thorough explanation of the inference stage for both traditional and open-vocabulary object detection tasks. While ground-truth boxes are used during training, it remains unclear whether bounding box information is utilized at inference. This raises questions about the potential impact of using predicted boxes and how the model maintains computational efficiency during inference.

2. The authors claim that “sensitivity to foreground objects is important,” yet there is no supporting reference or experimental validation for this assertion. Additionally, the motivation and intended effect of the Dynamic Query design for unseen categories are not clearly articulated, leaving its purpose somewhat ambiguous.

3. In the open-vocabulary setting, the OV-LVIS dataset is now widely adopted and is generally considered more challenging than OV-COCO. Providing additional results on OV-LVIS would strengthen the paper’s claim of improved performance under more challenging benchmark.

**Questions:**

See the weakness.

---

### Official Review · Reviewer_V2wV · 2024-11-02

**Soundness:** 3
**Presentation:** 3
**Contribution:** 3
**Rating:** 5
**Confidence:** 4

**Summary:**

In this paper, the authors propose two methods to enhance DETR-based detectors: a scale-aware contrastive approach to improve the encoder's feature representation, and a query encoding method to enhance the decoder's performance. The authors conduct extensive experiments on COCO, LVIS, and OV-COCO to demonstrate the effectiveness. However, it is important to note that the proposed methods are incremental, as similar improvement techniques have been widely adopted in previous works.

**Strengths:**

+ The proposed method is straightforward and easy to follow, with performance improvements across various detectors demonstrating its effectiveness.
+ The authors conduct experiments on a wide range of baseline detectors and downstream scenarios, showcasing the versatility of their approach. I appreciate the authors' effort across diverse settings.

**Weaknesses:**

- A major concern is that the proposed methods are incremental and lack academic novelty. The proposed AlignNet is similar to the RoIAlign mechanism used to enhance image feature maps in methods such as OV-DETR and RegionCLIP. Moreover, the Dynamic Query Learning approach is extremely similar to DN-DETR and DINO, differing mainly in query context encoding. This raises concerns that the proposed methods might be more akin to training tricks rather than novel techniques.
- The presentation could be improved. While the authors state that their motivation is to leverage the visual-linguistic capabilities of CLIP, they only use the CLIP text encoder for concept embedding. This approach raises the question of whether a language-only pre-trained text encoder, such as BERT, could be used instead. Additionally, there is a lack of analysis explaining why incorporating scale information into AlignNet leads to significantly better performance compared to using bounding boxes, as shown in Table 5.
- Some implementation details are missing. It is unclear which version of CLIP was used for the experiments and what the experimental setup was for the ablation studies.

**Questions:**

What was the experimental setup for the ablation studies? Which version of CLIP was used for the experiments? How does the performance vary when using different versions of CLIP or a language-only pre-trained model like BERT?

---

### Official Review · Reviewer_o6N3 · 2024-11-04

**Soundness:** 2
**Presentation:** 1
**Contribution:** 2
**Rating:** 5
**Confidence:** 3

**Summary:**

This paper introduces CLIP-DETR, an object detection framework that leverages the pre-trained visual-linguistic capabilities of CLIP to enhance both encoding and decoding in DETR-based models. By focusing on feature map sensitivity to objects and adaptable queries, CLIP-DETR addresses the limitations of prior approaches in object feature refinement, especially for unseen categories. Experiments show that CLIP-DETR outperforms state-of-the-art models in both standard and open-vocabulary object detection tasks.

**Strengths:**

- The proposed AlignNet module effectively leverages CLIP for category- and scale-aware feature refinement. This is a valuable idea with potential applications in other object detection frameworks and vision-language models (VLMs).

**Weaknesses:**

- The computational performance analysis of this paper is limited. Although the proposed modules are not required during inference, their impact on training time is unclear. Given that the performance increase is insignificant (+0.1 with SwinL), it is important to evaluate the trade-off here.

- This paper lacks a comparison with recent SOTA methods [1,2,3]. Additionally, its effectiveness on strong baselines could be explored further. For example, why did the authors not choose Co-DETR with a ViT-L backbone instead of Swin-L? This could potentially establish a new state of the art on the COCO benchmark.

[1] Minderer, Matthias, Alexey Gritsenko, and Neil Houlsby. "Scaling open-vocabulary object detection." Advances in Neural Information Processing Systems 36 (2024).

[2] Cheng, Tianheng, et al. "Yolo-world: Real-time open-vocabulary object detection." Proceedings of the IEEE/CVF Conference on Computer Vision and Pattern Recognition. 2024.

[3] Liu, Shilong, et al. "Grounding dino: Marrying dino with grounded pre-training for open-set object detection." arXiv preprint arXiv:2303.05499 (2023).

- The writing could be improved. Several claims in the introduction (lines 40–46) lack references. Moreover, the organization of the related work section, specifically Sec. 2.1, is somewhat disordered and difficult to follow.

Minor:
- The paper contains several typos and grammatical errors (e.g., lines 30–31, 246, 258).
- Tables 1 and 3 need more detailed captions. $\dagger$ is not explained in Table 1.

**Questions:**

- L187-188 states, "We aggregate the information by averaging the features across the L levels, producing a
single object encoded feature." In object detection, naive averaging of scale-level features often results in sub-optimal gains. What is the intuition behind this approach here? Is there an ablation study exploring alternative aggregation methods?

- Recent open-set/open-vocabulary object detection methods [1, 2, 3] are also capable of zero-shot detection. How does CLIP-DETR’s zero-shot detection capability compare to these methods?

[1] Minderer, Matthias, Alexey Gritsenko, and Neil Houlsby. "Scaling open-vocabulary object detection." Advances in Neural Information Processing Systems 36 (2024).

[2] Cheng, Tianheng, et al. "Yolo-world: Real-time open-vocabulary object detection." Proceedings of the IEEE/CVF Conference on Computer Vision and Pattern Recognition. 2024.

[3] Liu, Shilong, et al. "Grounding dino: Marrying dino with grounded pre-training for open-set object detection." arXiv preprint arXiv:2303.05499 (2023).

---

### Official Review · Reviewer_HNyB · 2024-11-05

**Soundness:** 2
**Presentation:** 2
**Contribution:** 2
**Rating:** 5
**Confidence:** 3

**Summary:**

To adapt image-level pretrained CLIP to object detection tasks, this paper proposes two modules to plug into DETR models, including object-text alignment and dynamic query mechanism. Their effectiveness is proved in close-set and open-vocabulary object detection experiment settings.

**Strengths:**

- The experiments on COCO and LVIS datasets show the effectiveness of proposed two designs.
- These two designs are only used during training, bringing no overhead to inference efficiency.

**Weaknesses:**

The proposed two designs help improve CLIP's abilities on detection tasks, however, I don't think the contribution of this paper is above the acceptance bar of ICLR main conference.

- Adding object-text alignment is a well-know knowledge when adopting CLIP to dense prediction tasks. Compared with recent two years papers in this direction, I don't learn new things from this paper, either from research insights or from engineering implementations.
- The proposed dynamic query is a very incremental improvement based on DINO.

**Questions:**

See weakness.

---

### Official Review · Reviewer_unJT · 2024-11-06

**Soundness:** 3
**Presentation:** 3
**Contribution:** 3
**Rating:** 8
**Confidence:** 3

**Summary:**

The paper proposes CLIP-DETR, an innovative object detection framework integrating CLIP's visual-linguistic capabilities into DETR-based models. It enhances both the encoding and decoding processes of object detection by aligning visual features with language models. The key contributions include AlignNet for feature refinement and a Dynamic Query Learning Mechanism (DynQL) for improving decoder robustness. The model showcases significant performance gains over state-of-the-art on both closed-set and open-vocabulary object detection tasks.

**Strengths:**

Innovation: The integration of CLIP's visual-linguistic features into DETR architecture addresses the adaptability and precision limitations of previous approaches. The novel AlignNet and DynQL mechanisms provide clear innovations in feature refinement and query optimization.
Empirical Performance: The paper provides comprehensive benchmarks showing that CLIP-DETR outperforms existing models on the COCO dataset and achieves notable improvements in open-vocabulary detection.
Detailed Analysis: The ablation studies are thorough, effectively demonstrating the individual and combined impacts of different components of the proposed model.

**Weaknesses:**

Complexity and Computation: The introduction of additional modules like AlignNet and DynQL might increase the model's complexity and computational requirements, which are not discussed in detail.
Legibility Issues: The text in Figure 1 is too small, making it difficult to read and understand the details of the model architecture and the data flow depicted.
Experimental Clarity: Some experimental settings, particularly for open-vocabulary tasks, are not clearly defined, which may lead to ambiguity in reproducibility and understanding the full scope of the model's capabilities.
Incomplete Open Vocabulary Experiments: The open-vocabulary detection experiments appear incomplete and lack a thorough exploration of the model's performance across a broader range of scenarios, as highlighted in the Questions section.

**Questions:**

Could the authors perform experiments using a naive DETR setup to compare its effectiveness against CLIP-DETR, particularly focusing on computational efficiency and detection accuracy?
Can the authors include experiments on the LVIS dataset for open-vocabulary object detection tasks to evaluate the model's performance in more diverse and challenging scenarios?
Could the authors provide detailed ablation studies on the open-vocabulary detection to assess the individual contributions of the novel components and the overall architecture's effectiveness in handling such tasks?
Can the authors elaborate on the training details of "Ours-CORA", specifically how AlignNet and DynQL are integrated and function within the two-stage training process? The current description does not clearly illustrate how these components contribute to the model's learning and performance across both stages.

---

### Meta-Review · Area_Chair_i3QS · 2024-12-09

**Metareview:**

The paper presents CLIP-DETR, an object detection framework that incorporates CLIP's visual-linguistic capabilities into DETR-based architectures. It achieves performance improvements in both closed-set and open-vocabulary detection tasks on datasets such as COCO and LVIS, with ablation studies highlighting the contributions of the proposed AlignNet and DynQL modules. While the paper offers practical contributions and measurable performance gains across various object detection scenarios, its novelty and clarity do not meet the high standard expected at ICLR. The proposed techniques are somewhat incremental, and the absence of comparisons with relevant baselines weakens the claims of improvement. Additionally, the authors did not adequately address the reviewers' concerns during the rebuttal period.

**Additional Comments On Reviewer Discussion:**

- Multiple reviewers (HNyB, V2wV, Rz2L) questioned the novelty of the proposed methods, noting their similarities to existing techniques.
- Several reviewers (o6N3, V2wV, Rz2L) noted the lack of comparisons with strong baselines like Co-DETR and more recent models.
- Reviewers (unJT, Rz2L) raised concerns about missing evaluations, particularly on OV-LVIS and zero-shot settings.
- Several reviewers (HNyB, o6N3, V2wV) noted unclear figures, missing implementation details, and disorganized sections.

No responses were provided to address any of these significant points during the rebuttal period.

---

### Decision · Program_Chairs · 2025-01-22

Reject